

# Niche differentiation in the rhizosphere and endosphere fungal microbiome of wild *Paris polyphylla* Sm.

Yan Wang[1,2,3], Hanping Wang[4], HuYin Cheng[5], Fan Chang[1,2], Yi Wan[2,3] and Xiaoping She[1]

[1] College of Life Sciences, Shaanxi Normal University, Xi'an, China
[2] Shaanxi Microbiology Institute, Xi'an, China
[3] Shaanxi Academy of Sciences, Engineering Center of QinLing Mountains Natural Products, Xi'an, Shaanxi, China
[4] College of Medical, Xi'an International University, Xi'an, China
[5] College of Pharmacy, Shaanxi University of Chinese Medicine, Xi'an, Shaanxi, China

## ABSTRACT

**Background:** The plant microbiome is one of the key determinants of plant health and metabolite production. The plant microbiome affects the plant's absorption of nutrient elements, improves plant tolerance to negative environmental factors, increases the accumulation of active components, and alters tissue texture. The microbial community is also important for the accumulation of secondary metabolites by plants. However, there are few studies on the niche differentiation of endophytic microorganisms of plants, especially at different elevations.

**Methods:** We investigated the effects of altitude on the community composition of endophytic fungal communities and the differentiation of endophytic microorganisms among different niches in *Paris polyphylla* Sm. The rhizosphere soil, roots, rhizomes and leaves of wild-type *P. polyphylla* Sm. at different altitudes were sampled, and the fungal communities of all samples were analyzed by internal transcribed spacer one amplification sequencing.

**Results:** The results showed that in rhizosphere soil, the number of operational taxonomic units (OTUs) that could be classified or identified decreased significantly with increasing altitude, whereas in the endosphere of plants, the total number of OTUs was higher at intermediate altitudes than other altitudes. Furthermore, the structural variability in the rhizosphere fungal community was significantly lower than that in the endophytic communities. In addition, our results confirmed the presence of niche differentiation among members of the endophytic microbial community. Finally, we also determined that the predominant genus of mycobiota in the rhizome was *Cadophora*. This study provides insight into the relationships between the endosphere microbiome and plants and can guide the artificial cultivation of this plant.

## INTRODUCTION

Interactions between microorganisms and plants have become a popular research topic in microbiology and botany, with studies conducted on tree (*Beckers et al., 2017*;

Corresponding author
Xiaoping She,
correspond2020@qq.com

*Cregger et al., 2018*), crop and floriculture microbiomes. Plants and microorganisms have close mutualistic or competitive relationships. In most cases, microorganisms play key roles in the survival and performance of plants (*Hackstein, 2010*; *Hacquard et al., 2015*). In addition, microorganisms are also important for the regulation of plant immune systems (*Jones & Dangl, 2006*; *Kau et al., 2011*; *Lebeis et al., 2015*; *Lee & Mazmanian, 2010*) and influence plant metabolism (*Khan et al., 2011a*, *2011b*). Microorganisms can influence medicinal plants in different ways, such as by affecting the absorption of nutrient elements, improving tolerance to stress, increasing the accumulation of plant active components, and causing changes in tissue texture. The presence of microorganisms in plants is also affected by the host plant genotype (*Cregger et al., 2018*). To some extent, microorganisms are specific to their hosts. Even within the same genus, endosphere microbiome communities may vary significantly.

The symbiosis between fungi and medicinal plants involves a series of complex processes, such as cell morphological changes, signal recognition, signal transduction, nutrient exchange and gene expression (*Huang, Long & Lam, 2018*). Endophytic fungi produce the same secondary metabolites as host plants (*Guo, 2016*; *Lu et al., 2018*; *Huang et al., 2007*; *Xue, 2013*). Some endophytic fungi can directly or indirectly affect the secondary metabolism of medicinal plants (*Yadav, Aggarwal & Singh, 2013*; *Bao et al., 2017*). In addition, the associated bacterial communities may play important roles in the regulation of the plant immune system (*Kau et al., 2011*; *Lebeis et al., 2015*). Therefore, endosphere microbiome communities are often called the second or extended genome of the host (*Beckers et al., 2017*). Recently, the effects of endosphere fungi on the growth, development, environmental stress resistance and secondary metabolite synthesis of medicinal plants have attracted increased concern.

Plant–Microbe interactions has received substantial attention in recent years as a subject of scientific and commercial interest (*Turner, James & Poole, 2013*). Studies have shown that the content of secondary metabolites from the same medicinal plant species can be different depending on their location of cultivation, which could in part be related to different composition in their associated microbes when grown at different sites (*Huang, Long & Lam, 2018*; *Köberl et al., 2013*). For some microbes, their metabolites could also be involved in modulating the production of bioactive phytometabolites, such as paclitaxel produced by *Taxomyces andreanae* (*Köberl et al., 2013*).

*Paris polyphylla* Sm. is a famous medicinal plant belonging to *Paris* in the Liliaceae family. This plant is found in the tropical and temperate regions of Eurasia and is mainly distributed in Southwest China around altitudes of 900–2,000 m. The rhizome is the medicinal part of the plant. Plant components reach the required effectiveness after 5 years. The chemical components of this plant have hemostatic and anti-inflammatory properties as well as good therapeutic effects in treating snake bites. In China, *P. polyphylla* Sm. is used as a raw material for many medicines. There are many studies on the chemical composition and application depth of this plant, but the endosphere microbiome is less understood. However, wild *P. polyphylla* Sm. is very rare. This study focuses on the endosphere microbiome of wild *P. polyphylla* Sm. The fungal communities of all samples were analyzed by internal transcribed spacer (ITS) one amplification sequencing.

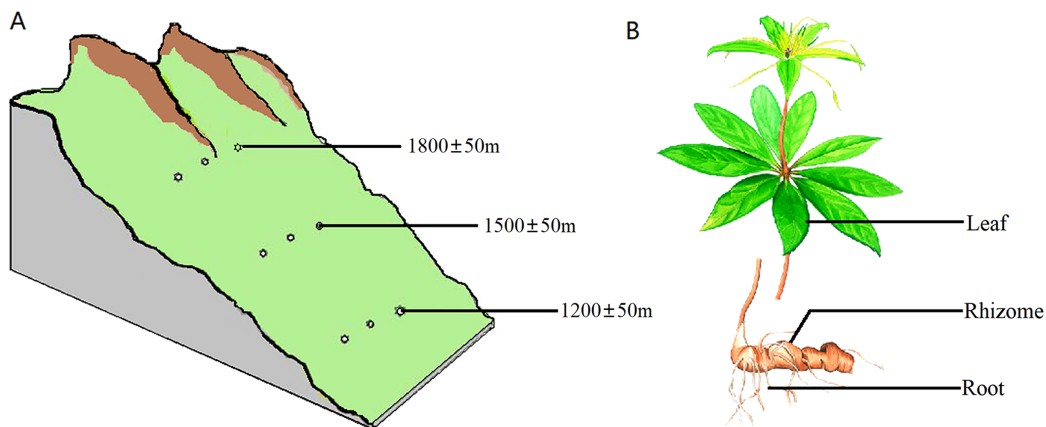

**Figure 1 Diagram of the sampling operation.** (A) Sampling locations. Sampling was carried out at three different heights, 1,200 ± 50 m, 1,500 ± 50 m and 1,800 ± 50 m, with three plants at each altitude at an interval of 300 m. (B) Sampled parts of the plant. For each plant, samples were separately collected in the rhizosphere soil, roots, rhizomes and leaves.

We assessed the fungal microbiomes in rhizosphere soil, rhizomes, and leaves of *P. polyphylla* Sm. at different altitudes; DNA samples were amplified by PCR and analyzed by sequencing. The niche differentiation of the related fungal microbiome was also investigated. These studies will help us understand the correlation between the endosphere microbiome and plants and help guide the artificial cultivation of plants.

## MATERIALS AND METHODS

### Field trials and sampling

The samples were collected from the southern slope of Qinling Mountains, Han Zhong, Shaanxi Province, China. The location was at a longitude of 107°41′~107°50′E and latitude of 32°16′~32°43′N. All samples were healthy and free of pests and diseases. All of the sampled trees were 4–5 years old based on bud scale scars. The sampling was carried out on plants with heights of 40–70 cm and on 8–11 leaves per plant. Sampling was carried out at three different altitudes, 1,200 ± 50 m, 1,500 ± 50 m and 1,800 ± 50 m, with three plants at each altitude spaced 300 m apart (Fig. 1A). Samples of the rhizosphere soil, roots, rhizomes and leaves were separately collected from each plant (Fig. 1B). Rhizosphere soil was strictly defined as soil particles adhering to roots. To standardize and maximize the reproducibility of plant samples, 1 g of roots and 5 g of rhizomes were collected from each plant. For leaf samples, the same number of leaves and petioles were removed from each group of plants.

### Sample processing

The samples were processed according to the method described by *Guo, Hyde & Liew (2000)*. Briefly, the excess soil was shaken from the roots, the soil that remained attached to the roots was defined as the rhizosphere soil. The root samples were separated from soil particles by shaking on a platform (10 min, 120 rpm/min). The soil particles that directly dislodged from roots represented the rhizosphere soil. The rhizosphere soil was

collected in an aseptic polyethylene tube and used for DNA extraction. The root, rhizome, and leaf samples were washed with flowing tap water to remove the excess soil; they were then rinsed with 75% alcohol for 3 min before being treated with 5% hypochloric acid for 2 min. Then, the plant samples were washed with sterile water three times to remove epiphytic microorganisms from the plant samples and obtain aseptic root, rhizome, and leaf samples. The rhizosphere soil, root rhizomes, and leaf samples were treated with liquid nitrogen (30 s) before the total DNA was extracted with the corresponding DNA extraction kit. All sample materials were stored at −80 °C before further treatment.

## DNA extraction

DNA was extracted from all the rhizosphere soil, root, rhizome and leaf samples. DNA extraction bias was minimized. Approximately 0.5 g of rhizosphere soil was used for each individual DNA extraction with the soil DNA extraction kit following the protocol provided by the manufacturer (MoBio Power Soil DNA extraction kit 12888-50). Approximately 0.5 g of plant tissue samples was used for each individual DNA extraction. The DNA of the plant tissue samples was extracted using the magnetic bead method (genomic DNA extraction kit DP342) according to the manufacturer's protocol.

## PCR amplification and sequencing

The internal transcribed spacer regions of the ribosomal RNA gene were amplified by PCR using the primers ITS1-1F-F CTTGGTCATTTAGAGGAAGTAA, ITS1-1F-R GCTGCGTTCTTCATCGATGC, ITS5-1737F GGAAGTAAAAGTCGTAACAAGG and ITS2-2043R GCTGCGTTCATCGATGC. PCR was carried out of a 20 µl mixture containing 4 µl of 5 × FastPfu buffer, 2 µl of 2.5 mm dNTPs, 0.8 µl of primer (5 µm), 0.4 µl of Fast Pfu polymerase and 10 ng of template DNA. The amplification products were extracted from 2% agarose gel, and the AxyPrep DNA gel extraction kit (Axygen Bioscience, United City, CA, USA) was used. Purifications were carried out according to the manufacturer's instructions and quantified by QuantiFluor-St (Promega, Durham, NC, USA).

The purified PCR products were measured by Qubit 3.0 (Life Invitrogen, Waltham, MA, USA), and the average mixture of each of the 24 amplified fragments with different bar codes was obtained. The Illumina library was constructed by using polymerized DNA products according to the preparation process of the Illumina genomic DNA library. The amplified library was paired and sequenced on the Illumina MiSeq platform (Beijing Novosource Bioinformation Technology Co., Ltd., Beijing, China) according to the standard protocol. The original data were stored in the National Center for Biotechnology Information (NCBI) sequence, and the archived (SRA: PRJNA504372) database is accessible via the link https://www.ncbi.nlm.nih.gov/sra/PRJNA504372.

## Sequence processing

Using the analysis platform at our research center, the raw sequence data were first screened by FastQC software to remove the low-quality sequences and then processed using USEARCH (version 11) (http://www.drive5.com/usearch/) for the

subsequent bioinformatics analysis. The USEARCH process was carried out using the fastq_mergepairs method to merge the two terminal sequences. Primer excision was analyzed using Cutadapt 1.18 (https://cutadapt.readthedocs.io/en/stable/). Usearchfastq_filter was used to control the sequence quality. The Fastx_uniques method was used to remove unnecessary and singleton sequences (the minimum parameter was 8). The unoise3 algorithm was used to cluster operational taxonomic units (OTUs) (sub-OTUs) without a reference, and the OTU table was generated by otutab. The clustered sequence was annotated by the Unite database, and the Syntax method had a cutoff value of 0.8. The OTU table was obtained after extraction using the least sequence value in the grouping.

## Statistical analysis

Statistical analyses were performed using R 3.5.1 (The R Foundation for Statistical Computing, Vienna, Austria) (*Edgar, 2010*). Alpha and beta diversities were analyzed using USEARCH. The ANOVA method and T-test method were used to study the α-diversity through global diversity on the niches of the rhizosphere soil, root, rhizome and leaf samples. Beta diversity, through the Binary Jaccard distance method, was analyzed by principal coordinate analysis (PCoA) and hierarchical clustering analysis of Unweighted Pair-group Method with Arithmetic Mean (UPGMA), and the distances between different niches were measured. A Venn diagram was used to display the numbers of common and unique OTUs and the shared OTUs among different samples (*Chen & Boutros, 2011*). The OTU sequence with the highest taxonomic abundance was selected as the representative sequence by Quantitative Insights Into Microbial Ecology (QIIME) software. Multiple sequence alignments were carried out, and the phylogenetic tree was constructed. The graph was prepared by using Python language.

# RESULTS

## Quality metrics of pyrosequencing analysis

A total of 6,159,204 raw reads were obtained by sequencing. The raw data were submitted to NCBI under the accession number PRJNA504372. After quality trimming and assigning the raw reads to the corresponding samples, 4,369,448 clean reads remained in the dataset. There were 121,373 clean reads per sample with an average length (±standard deviation) of 222 ± 50 bp (Table 1A). The readings that could not be clearly classified at the phylum level were also determined. These results showed that the number of OTUs that could not be classified was significantly higher in rhizosphere soil than in plants. The uncombined OTU number was lower in the plant compartments (roots, 16.68 ± 9%; rhizomes, 22.74 ± 30%; leaves, 28.23 ± 12%) than in the rhizosphere soils (46.16 ± 20%) (Table 1B). All unclassified data were removed from the dataset before further analysis.

## Alpha rarefaction curves and alpha diversity

Figure 2 shows the constructed alpha rarefaction curves. The change in the rarefaction curve was lower for the endosphere samples than for the rhizosphere samples, and the

**Table 1 Quality metrics of pyrosequencing analysis.** (A) Quality metrics before and after quality control (QC); the average read length was calculated based on 36 samples across all plant compartments. (B) Average number of assigned reads per plant compartment and undefined percentage of classified reads (±standard deviation). Each plant compartment was evaluated separately, and the data are from all samples in each mycobiota.

**A. Total number of reads and read length before and after quality checking and trimming**

| Total # of raw reads before QC | Average read length before QC | Total # of assigned reads after QC | Average read length after QC |
|---|---|---|---|
| 6159204 | 301 bp | 4369448 | 222 bp ±21 |

**B. Base and reads**

|  | Soil | Root | Rhizome | Leaf |
|---|---|---|---|---|
| Combined_base (bp) | 160245381 | 303613851 | 409652197 | 115621552 |
| Combined_reads | 665185 | 1437855 | 1712012 | 554396 |
| Average # of Combined reads | 73909 | 159762 | 190224 | 61600 |
| Uncombined (%) | 46.16 ± 20 | 16.68 ± 9 | 22.74 ± 30 | 28.23 ± 12 |

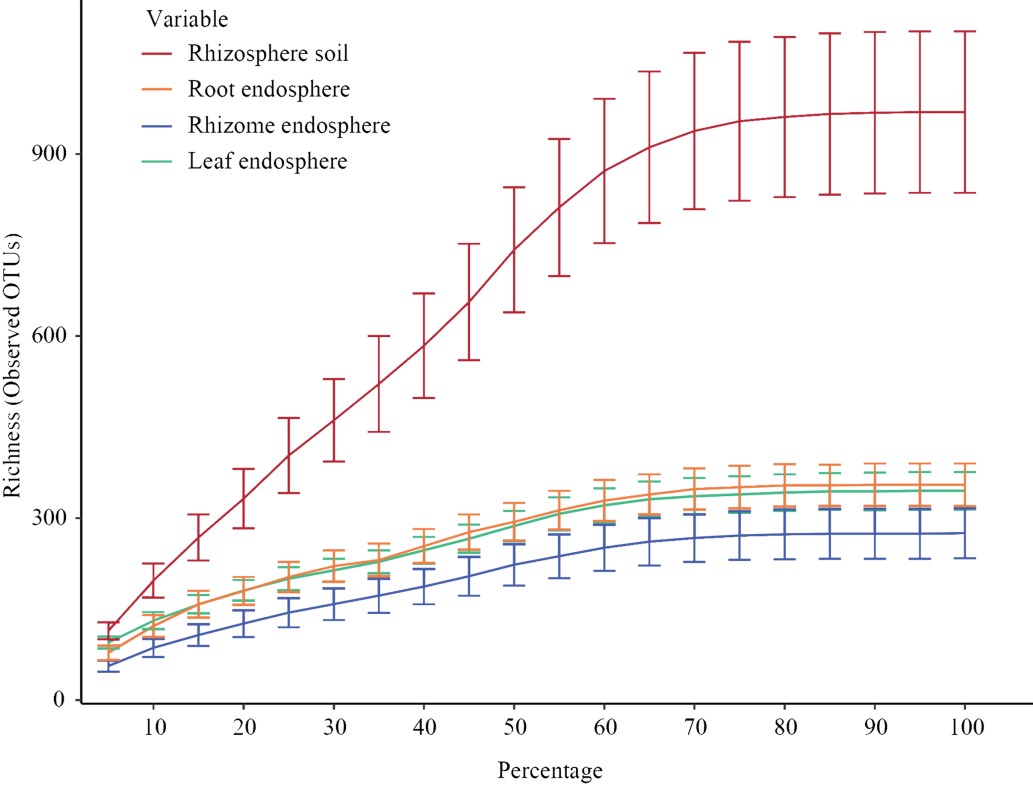

**Figure 2 Rarefaction curves of all samples for each group.** Rarefaction curves of all samples for each group (rhizosphere soil, roots, rhizomes, and leaves). Clustering was performed by unoise 3. Each curve represents the average of all repeats (±standard deviation) for each group of samples (rhizosphere soil, root, rhizome and leaf). The rarefaction curve of rhizosphere soil began to flatten at 70%, indicating that the depth of sequencing was sufficient to reliably describe the fungal communities associated with the plant departments.

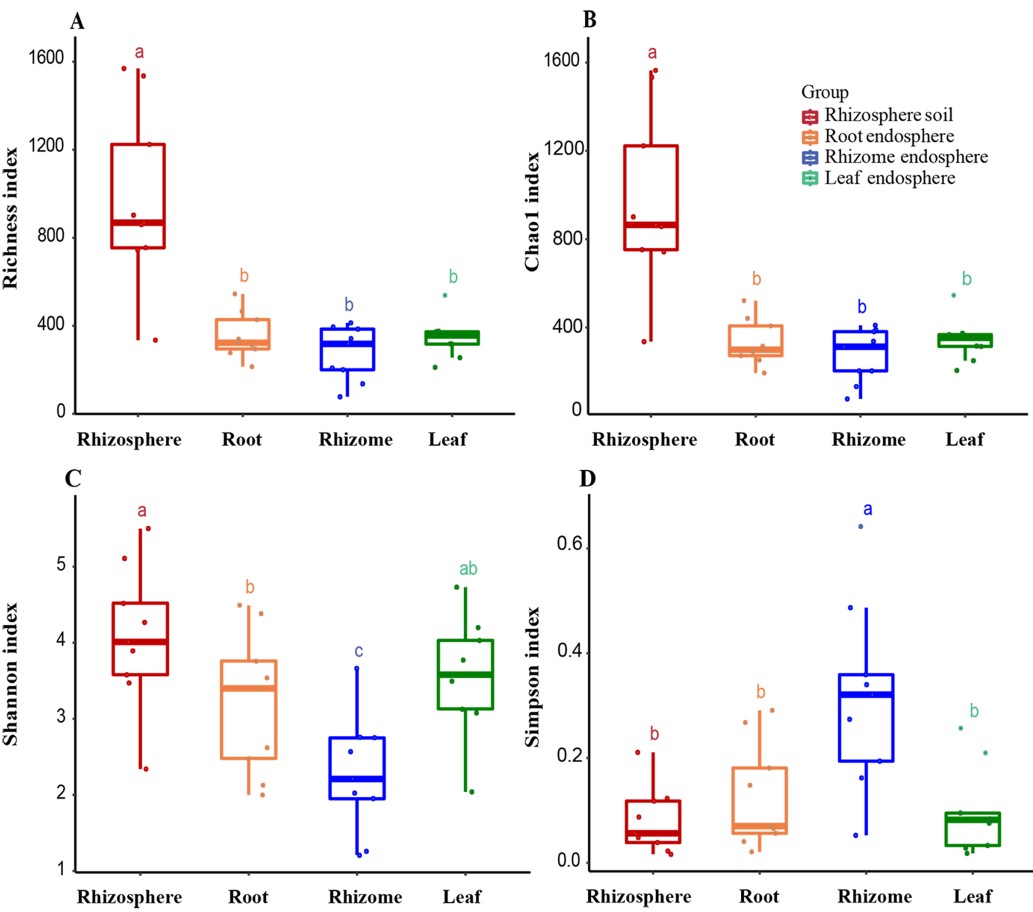

**Figure 3 Alpha diversity of the fungal communities in different samples.** (A) OTU richness estimates (number of observed OTUs). (B) Chao1 index. (C) Shannon index. (D) Simpson diversity index. The box diagram shows the first (25%) and third (75%) quartiles, median values and maximum and minimum observational values in each data set. The alpha diversity estimation is presented for the samples of rhizosphere soil, roots, rhizomes and leaves. The overall plant compartment $P$ value is displayed at the top of each graph. Significant differences ($P < 0.05$) across plant compartments are indicated with lowercase letters.

endosphere fungal diversity was much lower than the rhizosphere diversity. Clustering was performed by unoise3, and the rarefaction curves for evaluating OTU richness were close to saturation. Most endosphere fungi tended to saturate at 210~330 OTUs in the rhizomes and 300~380 OTUs in the roots and leaves. Rhizosphere fungi in the rhizosphere soil were saturated at 750~1150 OTUs. Statistical differences in OTU richness were inferred from alpha diversity measures (Fig. 3). The rarefaction curve of rhizosphere soil began to flatten at 70%, indicating that the depth of sequencing was sufficient to reliably describe the fungal communities associated with the plant compartments. The rhizosphere soil showed the highest fungal diversity, followed by the roots of *P. polyphylla* Sm., and the rhizomes had the lowest relative abundance of fungi, possibly due to rhizome selectiveness for endophyte colonization. Additionally, the accumulation of plant secondary metabolites may have affected the abundance of endosphere fungi. *Ding et al. (2018)* and *Wen et al. (2015)* studied secondary metabolites in different

compartments of *P. polyphylla* Sm. The types and contents of metabolites in different compartments of *P. polyphylla* Sm. are different.

## Alpha diversity

The analysis of the OTU richness, Chao1 index, Shannon index, and Simpson diversity index revealed the alpha diversity (microbial diversity) in each sample (Figs. 3A and 3B). The richness of rhizosphere soil was significantly higher than that of endosphere fungi (rhizosphere–root, $P = 0.000007$; rhizosphere–rhizome, $P = 0.000001$; rhizosphere–leaf, $P = 0.000005$). The richness of rhizosphere soil ($861 \pm 245$), roots ($323 \pm 70$) and rhizomes ($318 \pm 94$) showed a decreasing trend among different samples. The OTU richness index was slightly higher in the leaves ($356 \pm 29$) than in the rhizomes and roots. The Chao1 index also reflected a similar trend. According to the Shannon index and Simpson diversity index, the diversity of rhizosphere soil samples was higher than that of plant endosphere fungi, and the diversity of plant rhizomes was the lowest (Figs. 3C and 3D).

## Beta diversity

Beta diversity was analyzed at the OTU level, and the composition of fungal communities in different ecological niches were compared. The Binary Jaccard dissimilarity matrix was calculated using a PCoA to show the global similarity of fungal community structures in different samples (Fig. 4).

The PCoA showed that the rhizosphere soil communities were clustered at different phylogenetic levels, and no obvious clustering was found in the plant fungal communities. At the OTU level, principal component PC1 explained 27.74% of the total variation, and PC2 explained 15.99%. Rhizosphere soil samples were well separated from the root, rhizome and leaf samples (Fig. 4A). Hierarchical clustering of the samples was based on Binary Jaccard dissimilarity. Similarities based on Binary Jaccard (Fig. 4B) were superimposed on the PCoA plot. For the plant samples, the microbes were not entirely clustered according to the intervals among different plant compartments (Fig. 4). These situations indicated that the microbial communities were similar among different plant compartments. To statistically support the visual clustering of the fungal communities in the above PCoA analyses, different plant compartments were examined using ANOSIM (an analog of univariate ANOVA) with the Spearman rank correlation method (Table 2). All plant compartments exhibited fungal microbiota that were significantly dissimilar from each other ($P$ values listed in Table 2) at the OTU level.

## Phylum and genus level differences in the fungal microbiome in different habitats

We analyzed the classification of fungi at the phylum and genus levels. A species analysis was also carried out and showed that there were 8 phyla in rhizosphere soil, seven phyla in roots, seven phyla in rhizomes and seven phyla in leaves. The fungal communities were successfully distinguished among the different soil/plant parts (rhizosphere soil, roots, rhizomes and leaves) at the phylum level. The relative abundances of fungal taxa

A

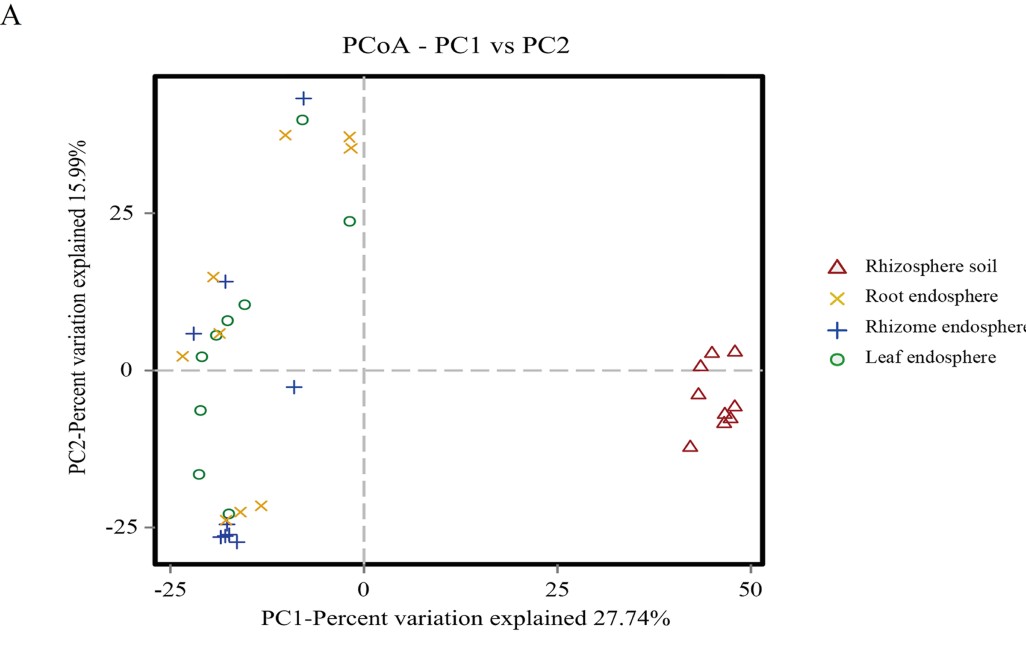

B

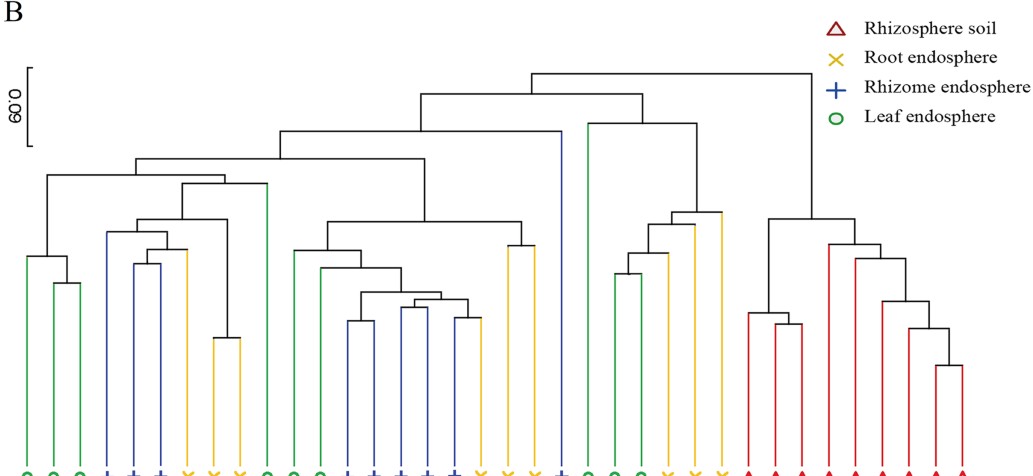

**Figure 4 Plant compartment drives the composition fungal communities at the OTU level.**
(A) The horizontal and vertical coordinates are the two eigenvalues that lead to the greatest difference between samples, and the main influence degree is reflected as a percentage. PCoA to show the global similarity of fungal community structures in different samples OTUs differentiating the plant compartents are displayed as vectors on the PCoA plots. Rhizosphere soil samples were well separated from the plant comparts samples. (B) Based on the PCoA of the Binary-Jaccard algorithm and hierarchical clustering analysis of Unweighted Pair-group Method with Arithmetic Mean (UPGMA), the similarity of species composition between samples, hierarchical clustering of the different samples based on Binary-Jaccard is determined. The closer the sample is, the shorter the branch length is, indicating that the species composition of the two samples is more similar. Similarities based on Binary-Jaccard were superimposed on the PCoA plot.

varied with plant compartment. The relative abundance of *Basidiomycota* was higher (26.19%) in the rhizosphere soil than in plant parts (roots 16.61%, rhizomes 10.16% and leaves 23.61%). The proportion of Ascomycota was higher in plants ($P \leq 0.01$) than in rhizosphere soil (15.35%). These results indicated that the plants had a selective

**Table 2 Analysis of similarity.** Plant compartment effects on the fungal community structures were calculated using ANOSIM (analysis of similarities) with the Spearman rank correlation method. Plant compartments (soil, root, stem, leaf) were a priori defined groups at OTU phylogenetic levels.

| Phylogenetic level ANOSIM output | R | P |
| --- | --- | --- |
| Soil vs. Root | 0.997 | 0.001** |
| Soil vs. Rhizome | 0.940 | 0.001** |
| Soil vs. Leaf | 0.987 | 0.001** |
| Root vs. Rhizome vs Leaf | 0.06 | 0.915 |

Notes:
** $P \leq 0.001$.
ANOSIM test statistic.

enrichment preference for Ascomycota. The abundance of *Ascomycota* was also higher in rhizomes (85.89%) than in other plant parts, possibly because rhizomes are the primary part of the plant that produces medicinal components. The role of Ascomycota in the synthesis of active components needs further study. *Glomeromycota* also occupied a certain proportion in the parts of the plants (roots 12.14%, rhizomes 2.82% and leaves 3.50%), while its enrichment in soil was only 0.01%. This result indicated that the plant parts had a preference for *Glomeromycota*.

At the genus level, 72 genera were annotated in rhizosphere soils, 75 in roots, 80 in rhizomes, and 78 in leaves. The distribution of the top 10 genera was also different among the rhizosphere soil, roots, rhizomes and leaves. In the rhizosphere soil, except in the unassigned soil, *Clavulina* was found to be the most abundant genus (11.29%). The enrichment of *Clavulina* was less than 1% in both plant parts (roots 0.07%, rhizomes 0.002% and leaves 0.03%) and rhizosphere soil. The enrichment of *Cadophora* was less than 1% in rhizosphere soil but much higher in plant parts (roots, 6.06%; rhizomes, 24.84% and leaves, 2.53%). The relative abundance of *Cadophora* was in the order of rhizome > root > leaf in the plant compartments, which was consistent with the results at the phylogenetic level; *Cadophora* belongs to Ascomycota. *Ascomycota* is considered the most likely endophytic taxon to colonize plants (*Guo, 2016*). *Ascomycota* is widely found in soils and in plants such as forage grass, flowers and crops (*Hacquard et al., 2015*, *Edwards et al., 2015*).

In addition, the distribution of genera and phyla of the different fungal communities (rhizosphere soil, roots, rhizomes and leaves) among different altitudes was analyzed. As shown in Fig. 5B, at the phylum level, the fungal phyla at different altitudes showed different trends. In the rhizosphere soil, the number of unassigned OTUs increased gradually with increasing altitude, which is consistent with previously reported results (*Guo, Hyde & Liew, 2000*). In addition, the microbiome also varied among different plant compartments. High enrichment of *Ascomycota* was found in the plants at the low-altitude areas, especially in the roots (91.42%) and leaves (82.54%). However, in the rhizomes of plants, the relative abundance of *Ascomycota* was higher (95.09%) in the middle-altitude area than in the other areas. *Basidiomycota*, another important phylum in plants, also varied with the altitude gradient. In the roots and leaves, *Basidiomycota* showed an

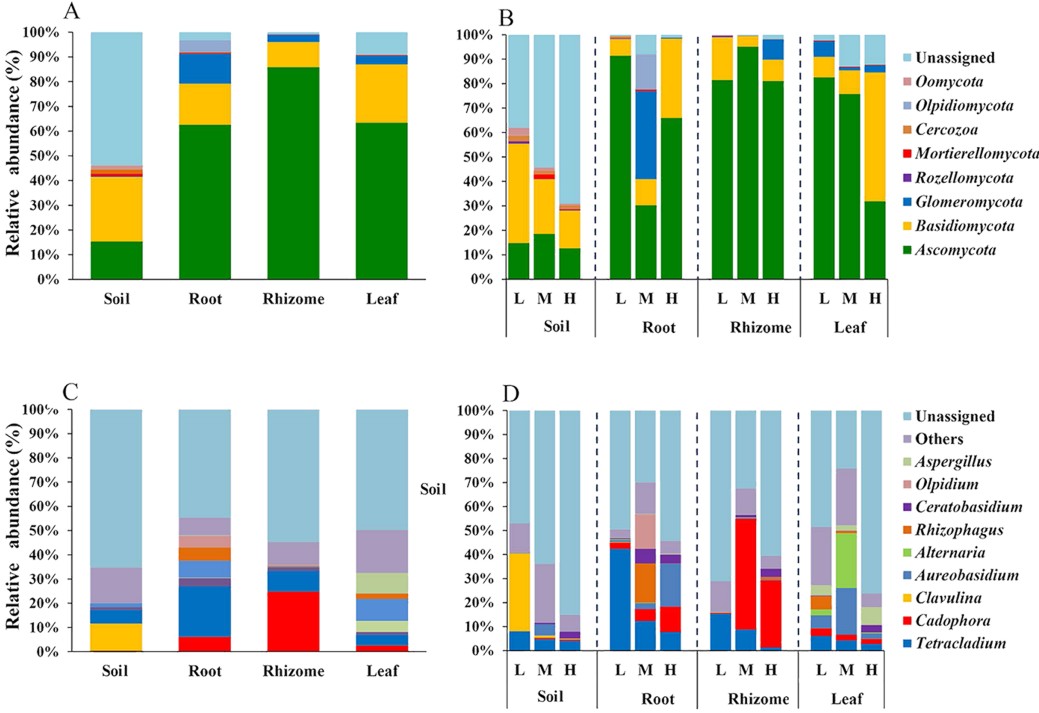

**Figure 5 Distribution of OTUs at the phylum and genus levels.** (A) The fungi phyla at different niches. (B) The fungi phyla at different altitudes show different trends at the phylum level. (C) The fungal genus at different niches. (D) The fungal genus at different altitudes showed different trends at the genus level. For the best display, only the top 10 species of abundance are displayed, and the other species are merged into "Others". Unclassified represents the species that have not been annotated by taxonomy. Major contributing phyla are displayed in different colors, and the unannotated are grouped and displayed in grayish blue. L, low altitude 1,200 m. M, middle altitude 1,500 m. H, high altitude 1,800 m. Coverage at the phylum levels: The percentages of the total community covered by the core OTUs were 46.16% (rhizosphere soil), 96.76% (roots), 99.19% (rhizomes) and 90.86% (leaves). Coverage at the genus level: The percentages of the total community covered by the core OTUs were 40.34% (rhizosphere soil), 52.20% (roots), 47.17% (rhizomes) and 56.66% (leaves).

increasing trend with increasing altitude, but such a trend was not found in the rhizomes of plants. The fungi also showed different trends at the genus level (Fig. 5D). In rhizosphere soil, the annotated OTUs showed a downward trend with increasing elevation to 1,800 m (only 14.95%). The dominant mycobiota, such as *Tetracladium* and *Clavulina*, also showed decreasing trends with increasing elevation. The number of annotated OTUs in each plant was higher in the middle-altitude area than in the low and high-altitude areas (roots 70.07%, rhizomes 67.63% and leaves 75.85%). *Tetracladium* also decreased in abundance with increasing altitude. *Cadophora*, with a high enrichment rate in plants, showed different trends in different flora. The enrichment of *Cadophora* in rhizomes was greatest at the middle altitude (46.17%). To obtain a complete overview of OTU distribution in different plant compartments, we calculated the OTUs in each particular plant compartment as well as the OTUs shared among the different plant compartments (Fig. 6). R software was used to draw a Venn diagram (*Caporaso et al., 2010*). The proportion of exclusive OTUs in rhizosphere soil was 22.89%, and the

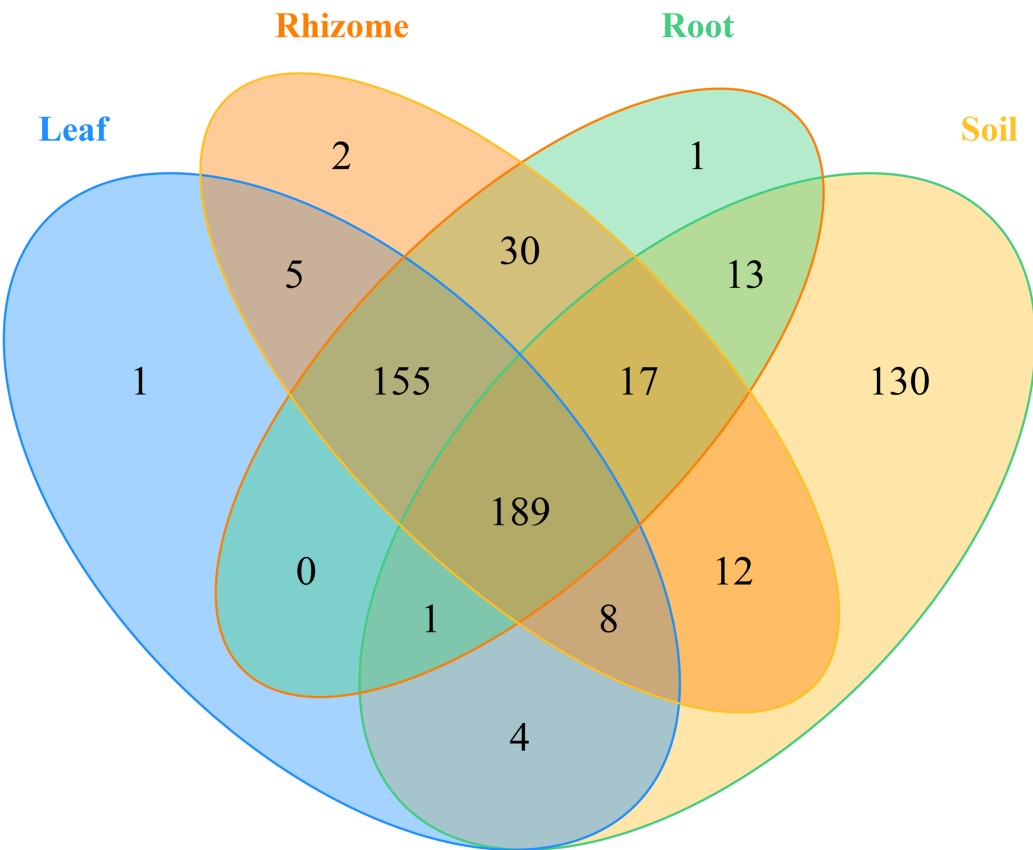

**Figure 6 Venn diagrams of the number of OTUs in different mycobiota.** The numbers within overlapping regions of ellipses are the total numbers of OTUs shared among the samples, and the numbers of unique OTUs in each sample are presented in the non-overlapping regions. The proportion of exclusive OTU in the soil was 22.89%, and the proportion of OTU in the plants was 34.15%. The OTUs simultaneously detected in soil and plants was 42.96%.

proportion of OTUs in plants was 34.15% (0.18% in the roots, 0.35% in the rhizomes and 0.18% in the leaves). The overlap rate of OTUs between rhizosphere soil and the root and rhizome samples was 38.73% and 39.79%, respectively. The overlap rate between rhizosphere soil and leaf samples was 35.56%. The proportion of OTUs simultaneously detected in rhizosphere soil and plants was 42.96%.

 The annotated microbial communities in the rhizosphere soil were mainly composed of *Clavulina*, *Tetracladium*, *Ceratobasidium* and *Aureobasidium*. These genera and *Rhizophagus* constituted the dominant microorganism community in plants, with extremely varied ecological niches. The dominant genera in the root samples were *Tetracladium*, *Aureobasidium* and *Cadophora*, but the dominant genera in the rhizome samples were *Cadophora*, *Tetracladium* and *Ceratobasidium*. The leaf samples mainly contained *Aureobasidium*, *Alternaria*, *Kluyveromyces*, *Cladosporium* and *Tetracladium*. These microbes have been isolated from a large variety of plants and have considerable promoting effects on plant health and growth (*Raven, 1970*; *Gottel et al., 2011*). *Cadophora* and *Tetracladium* were found to effectively colonize and enrich *P. polyphylla* Sm., with
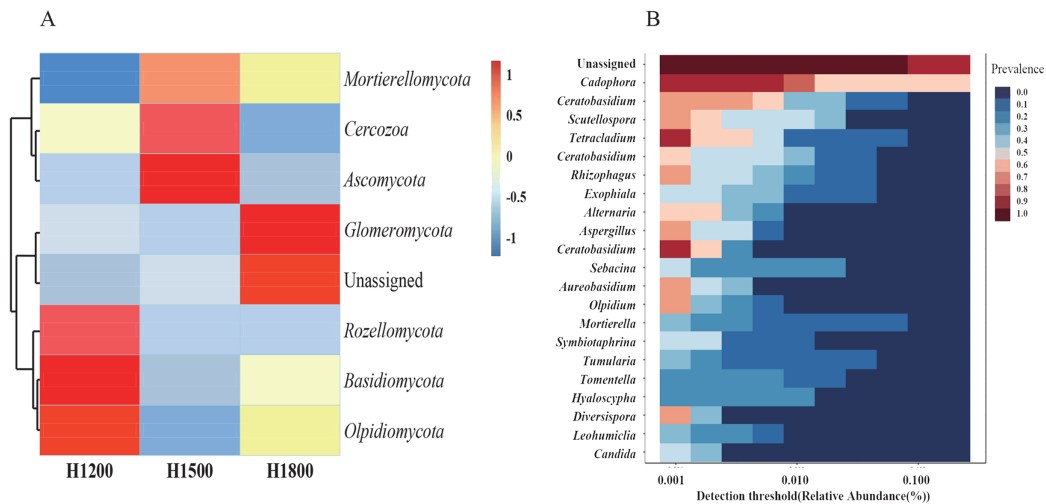

**Figure 7  Cluster heat maps of phylum and genus abundances.** (A) The abundances of different phyla at each elevation are shown. The horizontal cluster is the sample information at different altitudes; the longitudinal cluster is the species information; the left cluster tree is the species clustering tree; the right side contains the information of the species belonging to the phylum. (B) Heat map representing the core microbiome at the genus level. The different colors indicate the prevalence of the corresponding genera: red, high prevalence; blue, low prevalence. *Cadophora* was the core mycobiota type in the rhizome.

significant differences in relative abundance in roots (6.06% and 20.82%, respectively), rhizomes (24.84% and 8.46%, respectively) and leaves (2.53% and 4.39%, respectively).

## Cluster analysis of microbiome abundance in rhizome

As the rhizome is the medicinal part of the plant, we analyzed the species abundance of microbes in the rhizomes. The medicinal parts of wild rhizomes were analyzed at different altitudes. Cluster heat maps of species (Fig. 7A) were developed on the basis of species composition and the relative abundance of each sample. All data of the samples at the same altitude were combined and mapped using the R tool. The core mycobiota at the genus level were also analyzed (Fig. 7B). *Cadophora* was the core fungal microbiome in the rhizome.

After analysis with QIIME software, the sequence of OTUs for the most abundant genus in the rhizome compartment was selected as the reference sequence to perform multiple sequence alignments and construct the phylogenetic tree (Fig. 8). We defined five phyla with high species abundance in the rhizomes. Ascomycota formed a large independent branch, and its species abundance presented clear advantages over other phyla. This result further confirms the floristic selectivity of the host.

## DISCUSSION

### Quality of the pyrosequencing analysis

We used ITS1-1F and ITS1-5F primer mixtures to maximize the phylogenetic coverage of fungi. The high abundance of chloroplast genes can lead to unexpected co-amplification of non-target sequences (*Kabir, Peter & Jennifer, 2016*; *Roesch et al., 2007*). In this

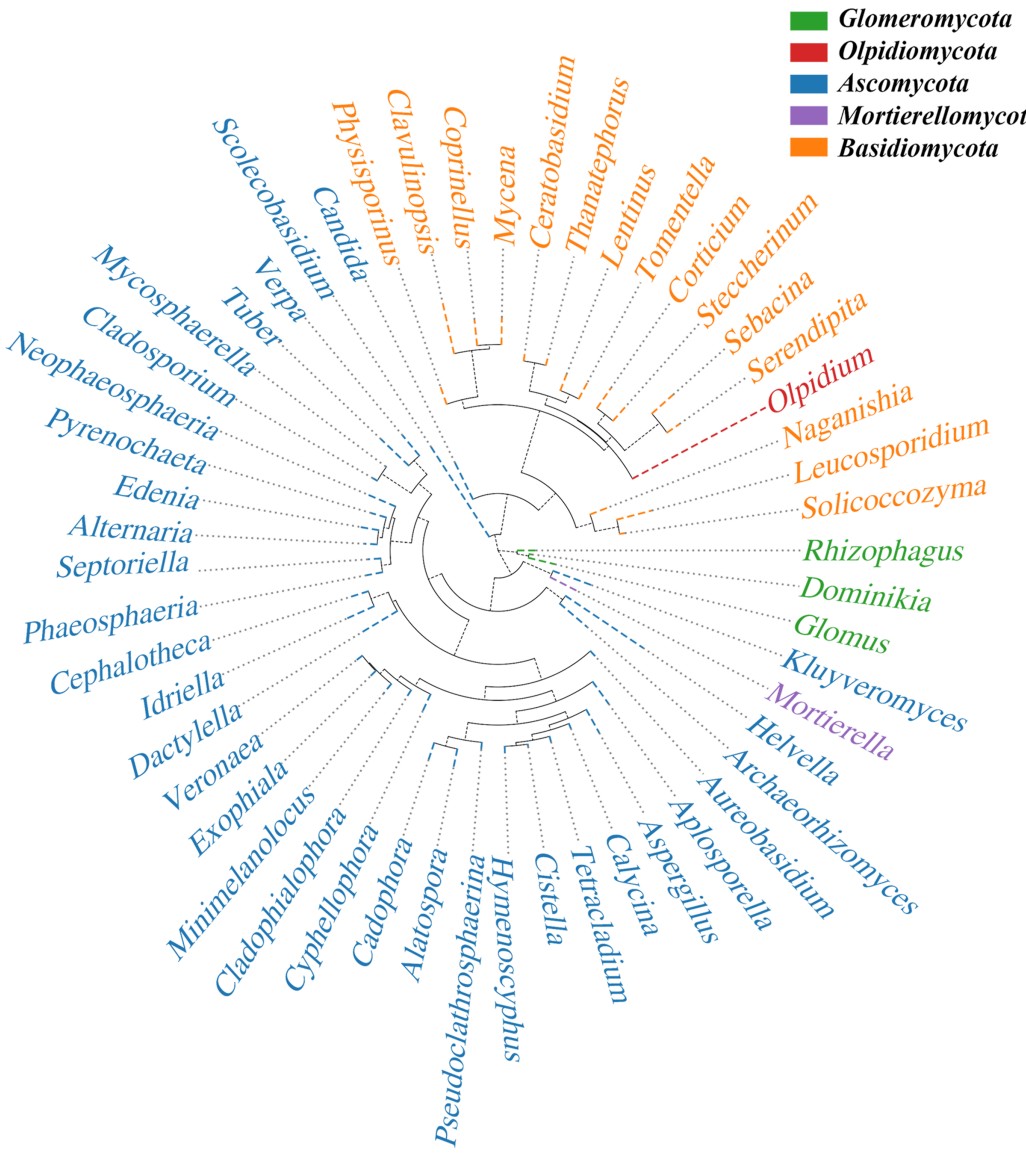

**Figure 8 Phylogenetic tree.** The same color name represents the same phylum. The phylogenetic tree was constructed using the Python tool. Each branch in the tree represents a species, and the length of the branch is the evolutionary distance between two species, namely, the degree of difference between the species.

study, the amplified plant chloroplast genes were culled. Remarkably, the singletons in rhizosphere soil and in different plant compartments had been removed before annotation. Due to the structural differences between soil and plants, we were unable to extract high-quality and high quantity DNA from all plant samples using the same DNA extraction kit. To ensure high-quality and high quantity DNA from all studied samples, we chose different pre-processing methods, and extraction kits were used for DNA extraction from rhizosphere soil and plant samples. As a result, the high discrepancy in the number of singletons in the plant parts could be attributable to genuinely rare (singleton) OTUs in the rhizosphere soil (Table 1). In fact, microbiomes in the rhizosphere soil

are generally considered the most diverse regions (*Lugtenberg & Kamilova, 2009*; *Coleman-Derr et al., 2016*). For further analysis, we chose to remove all singletons from the data sets. However, the involvement role of singletons ecological roles functions is largely unknown, this requires need the further study.

## Endophytic fungal communities in different niches

We estimated the richness, evenness and diversity of alpha diversity based on OTUs. We found that the abundances of OTUs in rhizosphere soil and plant compartments (roots, rhizomes and leaves) were clearly different. The results are consistent with the general view of endosphere colonization. The diversity and evenness varied greatly from rhizosphere soil to endosphere. Because of chemotaxis and colonization of the rhizosphere microbiome, rhizosphere microbial communities are rich and diverse (*Huang, Long & Lam, 2018*; *Bulgarelli et al., 2012*; *Hardoim, Van Overbeek & Van Elsas, 2008*). The rhizosphere soil-root interface acts as a selective barrier for endosphere fungal colonization. High variability of endophytic OTU richness, as depicted by the box plot, could possibly be caused by sporadic and nonuniform colonization of the roots, rhizomes and leaves of Paris (*Gottel et al., 2011*; *Beckers et al., 2017*). Therefore, our data suggest considerable variation in endophytic colonization as a major reason for the high variability in the box plot. At the genus level, *Cadophora* and *Tetracladium* were the dominant microorganisms among the plant mycobiota. Although their OTU abundance was abundances were different, both *Cadophora* and *Tetracladium* belong to *Ascomycota*, which is considered the most likely endophytic taxon to colonize plants (*Guo, 2016*). *Ascomycota* is widespread in soil and plants, such as forage, flowers and crops (*Kottke et al., 2008*; *Edwards et al., 2015*; *Hacquard et al., 2015*). Enrichment and depletion of specific microbiomes within the plant-associated microbiome are initiative processes that depend on active selection of microbial consortia by the plant host and opportunistic colonization of the available ecological niches (*Bulgarelli et al., 2013*; *Mehta & Rosato, 2001*; *Wen et al., 2007*). Therefore, the colonized fungi are limited to specific fungal species. Our results illustrated that the diversity and evenness decreased from the rhizosphere to the endosphere. Only a limited number of microbes could adapt to the way of life in the plant. As a result, the plant niches consist of specific endosphere communities. Previous studies of other plants have also reported niche differences in the distribution of endogenous microbial communities, such as poplar (*Beckers et al., 2017*; *Cregger et al., 2018*), which may have an effect on plant metabolism (*Chen et al., 2018*). Different niches of the plant may be associated with differences in plant metabolism. Studies of the differences in the endogenous microbial community in different niches could provide a basis for future studies of differences in metabolism in different parts of medicinal plants.

To compare the structure of the endosphere fungal community in different plant compartments, we clustered all samples utilizing PCoA and hierarchical clustering (Binary-Jaccard) (Fig. 4). At the OTU level, the rhizosphere soil gathered together. However, there was no obvious relationship between the fungal microorganism clusters in different plant tissues. This result was further confirmed by UPGMA hierarchical cluster analysis. *P. polyphylla* Sm. is a perennial herb. In each growth cycle of *P. polyphylla*

Sm., only the aboveground part of the plant dies, while the rhizome remains alive. This growth pattern may be one reason for the lack of obvious typical mycobiota in different plant compartments. In addition, for microorganisms, the endophytic environment of plants is complex, In contrast to rhizosphere colonization, intricate interplay between endophytic microbe and the host plants innate immune system, it of the host plant is completely different from the soil, which is also an important reason for the difference between the soil and plant microbial communities.

In this study, different rarefaction curves (Fig. 2) were obtained from the rhizosphere soil and endosphere samples. Compared with plants, the rhizosphere soil displayed much greater microbial diversity, possibly because the plant provides a relatively stable interior environment, which leads to lower variability in the plant's internal fungal community. In addition, we found that the OTU numbers in soil were almost three times those in plants. This result is consistent with the widely accepted view that soil contains many microbes. The structure of the endosphere fungal communities varied more markedly than the structure of the rhizosphere community (*Bulgarelli et al., 2012*; *Nallanchakravarthula et al., 2014*). Soil microbe communities form one of the most abundant microbial ecosystems on Earth (*Coleman-Derr et al., 2016*; *Wu et al., 2005*). In addition, root exudates and nutrients from mucilage sources also attract countless organisms to gather in the rhizosphere. The microbes in plants need to be strongly competitive to successfully colonize roots (*Hackstein, 2010*); competitive ability might manifest as the ability to break through plant cells (producing enzymes that degrade cell walls) and the ability to adapt to the innate immunity of plants (*Turner, James & Poole, 2013*; *Compant, Clément & Sessitsch, 2010*; *Nie et al., 2005*).

## Differences in the microbiome of plants among different altitudes

The content of secondary metabolites from the same medicinal plant species can vary depending on the location of cultivation, which could in part be related to differences in the composition of the associated microbial communities at different sites. According to *Köberl et al. (2013)*, many of the variations in the quality of traditional herbal medicine may be attributable to changes in the microbial community either in the rhizosphere or in the endophytic compartment of the medicinal plant (*Huang, Long & Lam, 2018*). The functional characteristics of endophytic communities residing inside the roots of rice have revealed that endophytes may be involved in the metabolic processes of rice (*Vain et al., 2014*). *Salvia miltiorrhiza* harbors a distinctive microbiome that is enriched in functions related to secondary metabolism and thus may contribute additional metabolic capabilities beyond those encoded in the genome of the host plant (*Huang, Long & Lam, 2018*; *Chen et al., 2018*). These observations suggest that the soil and climate at different locales can influence the metabolite content of the medicinal plant. Altitude is one of the most important factors affecting climate, therefore we analyzed the variation in the endophytic fungal communities among different altitudes. We divided the plant distribution into three elevation zones: low, middle and high altitude (Fig. 5). The results showed downward trends of the richness and diversity of soil microorganisms in the rhizosphere with increasing altitude. Previous studies have shown that fungal community

composition is influenced by both ecological factors and evolutionary factors. Spatial scale is considered a major factor contributing to differences in fungal diversity (*Kabir, Peter & Jennifer, 2016*). However, the richness and diversity of endophytic microorganisms do not follow the distribution law of soil microorganisms. In the endophytic environment, the richness and diversity of endophytic microorganisms are the most abundant in the plant endophytic environments, which is similar to the distribution law of plants, rather than following the law of altitude. This phenomenon has rarely been reported in previous studies. Through systematic study of the associated microbiome in medicinal plants, we should be able to clarify the distribute situation of various microbes in plants at different elevations. This information could guide better selection of growing environments for the cultivation of medicinal plants, which in turn may improve the medicinal quality and evaluation standards of medicines that will facilitate their passing more rigorous scientific and commercial evaluations.

# CONCLUSIONS

This study revealed the structural variability and niche differentiation in the rhizosphere and endosphere fungal microbiomes of wild *Paris* plants at different altitudes. The results show that the structural variability in microbiome communities in the rhizosphere soil is lower in wild *P. polyphylla* Sm. than in endosphere fungal communities. The formation of rhizosphere fungal communities is a stable process, and endophytic colonization is variable. In addition, our data confirm reports of niche differentiation in rhizosphere soil-root microbiome communities. Furthermore, our study not only reveals the relationships and differences in endosphere fungal communities in various plant tissues but also clearly shows relationships between altitude and the endosphere microbiome in plants. With increasing altitude, the diversity of the plant endosphere microbiome first increased and then decreased; this pattern is inconsistent with the relationship generally observed between soil microorganisms and altitude. In addition, the core members of the endophytic microbial communities in the rhizome of *P. polyphylla* Sm. were successfully identified. The present findings provide a basis for further studies on the interactions between the endosphere microbiome and hosts and can inform efforts involving the artificial planting of this plant.

# ACKNOWLEDGEMENTS

The authors would like to thank ShiQiang Cheng, MingZhong Hao, QiLing Sun and DanNi Zhu for their assistance with the field work.

## Funding

This work was funded by the Science and Technology Program of Shaanxi Academy of Science through research project number 2016k-18. Furthermore, this work was financially supported by the Science and Technology Research Project of Shaanxi Province

Academy of Sciences Project 2018nk-01. The funders had no role in study design, data collection and analysis, decision to publish, or preparation of the manuscript.

## Grant Disclosures
The following grant information was disclosed by the authors:
Science and Technology Program of Shaanxi Academy of Science: 2016k-18.
Science and Technology Research Project of Shaanxi Province Academy of Sciences Project: 2018nk-01.

## Competing Interests
The authors declare that they have no competing interests.

## Author Contributions
- Yan Wang conceived and designed the experiments, performed the experiments, analyzed the data, prepared figures and/or tables, authored or reviewed drafts of the paper, and approved the final draft.
- Hanping Wang conceived and designed the experiments, prepared figures and/or tables, and approved the final draft.
- HuYin Cheng performed the experiments, analyzed the data, authored or reviewed drafts of the paper, and approved the final draft.
- Fan Chang performed the experiments, analyzed the data, prepared figures and/or tables, and approved the final draft.
- Yi Wan conceived and designed the experiments, performed the experiments, analyzed the data, authored or reviewed drafts of the paper, and approved the final draft.
- Xiaoping She conceived and designed the experiments, prepared figures and/or tables, authored or reviewed drafts of the paper, and approved the final draft.

## Data Availability
Data is available at NCBI under the accession number PRJNA504372.

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
