# Peer review of "Niche differentiation in the rhizosphere and endosphere fungal microbiome of wild Paris polyphylla Sm"

_PeerJ, doi:10.7717/peerj.8510_

## Round 0.1 · original submission · Major Revisions

Please pay attention to the suggestions by reviewer 1 in particular

·

Basic reporting

The authors used sound statistics to support their work. The background is sufficient enough for the readers to understand the concept of why this work was performed. But I have some reservations. Probably reference list should include some more. The figure legends need to be fixed. There are certain ambiguous statements that needs to be corrected.

Experimental design

The question was well defined. But had it been the authors had some phytochemical data that would have made this work interesting. Experimental methods and statistical procedures were done rigorously.

Validity of the findings

The authors need to expand more on their results in their discussions. The data should have been supported by some phytochemical data so as to enrich.

Additional comments

It was interesting to go through this manuscript. The experiment was performed with rigorous statistical procedures and bioinformatic analysis. But I have certain reservations.
1.The authors should reflect more in the discussion.
2. Introduction can be expanded a bit.
3. I would suggest the authors to write the identified bacteria with a word 'ÓTU' besides the name. As it is an identified Operational Taxonomic Unit (OTU) .
4. Where ever the scientific name of the plant or the name of bacterial OTU (whether it is genus or species) was written, it should be written in the standard format (e.g. italics) for example in the line 26 and 27
5. The samples name or list in the figures should be same, as in some cases it is written as Rhsoil, Soil, Rhizosphere soil.

Line 26 Remove the extra dot after 'Sm'.
Line 50 Space should be added after 'genotype'
Line 84 Correct the formation of sentence
Line 86: This sentence is confusing when you compare it with line 84. At one place you are saying that you have collected the samples separately and another place you were saying complete plants. Rephrase it.
Line 92: Rephrase this sentence 'the root soil was shaken
Line 106: I could not understand, what the authors want to say regarding these two phrases?
1. 'the deviation of DNA extraction
2. 'Extracted of DNA with unified operate'.
Line 115: What is meant by '3 parts', Were the authors referring to three replicates ?
Line 135: Did you remove the sequences less than 300bp or the terminal primer sequences or the adapters as it is confusing?
Line 160: In the line 152, it says that using Primer deletion software you have removed the sequences that are less than 300 bp. Then how could it be that you used 272 bp (Max) and 172 (min) were used for the analysis. Can you please clarify it?
Line 201: Hardly 40% of the variation is being explained. So how can you say that it is significant. You need to have some kind of data that supports this data statistically and as well as significantly.
Line 207: when you have already classified at Phyla and Species level. Why was Species analysis was carried out to identify phyla?
Line 224: It should be leaves of leaf.
Line 231: Re-write the sentence as it is confusing
Line 290: Are you sure? because Bodenhausen et al 2013 work was about bacterial communities using 16S region primers (V5-V7 region). But you are talking about the fungi. In general, plant chloroplast always gets amplified during bacterial DNA amplifications.
Line 309: What is it meant by 'OTU levels were different'. Were you referring to the relative abundance?
Line 310: Rewrite the sentence.
Line 313 to 316: Please rewrite the sentence as it is confusing.
Line 333: Many articles have also published this information such as Bulgarelli et al 2012, Lundberg et al 2012 with respect to bacteria and Nallanchakravarthula et al 2014 with respect to fungi. It is better to add those references also.
Line 346: instead of fungal write it as 'fungi' or do you want to add another word after 'fungi'?

Reviewer 2 ·

Basic reporting

The paper describes the fungal community of Paris polyphylla Sm. tissues. Although the authors found interesting results regarding fungal diversity and composition, the paper is merely descriptive and authors should consider further studies to test their hypothesis. Also, English needs substantial improvement and the text needs a revision since there are several typing mistakes.

Experimental design

No comment.

Validity of the findings

The authors should consider how they present their results, such as:
- Why use Chao1, Shannon, and Simpson to describe alpha-diversity if you are not discussing trends that are shown in them?
- Fig 5 should show all the replicates to demonstrate that the communities are consistent in each microbial assessment.

Additional comments

No comment.

---

## Round 0.2 · Minor Revisions

After consultation with the Section Editors, we would like you to revise the manuscript to explicitly state the hypothesis/hypotheses (or state the null hypothesis/ses).

Thank you

·

Basic reporting

no comment

Experimental design

no comment

Validity of the findings

no comment

Additional comments

I appreciate the authors for accepting the suggestions of the reviewers and improved the manuscript. I hope the authors will do the future experiments keeping in mind the suggestions made the reviewers.

Reviewer 2 ·

Basic reporting

The manuscript "Niche Differentiation in the Rhizosphere and Endosphere Fungal Microbiome of Wild Paris polyphylla Sm." describes the fungal community of Paris polyphylla Sm. in different altitudes.

Experimental design

Why the authors sampled the community at different altitudes? Why is this important to this medicinal plant? And regarding the differents found in the fungal community at different altitudes, what is the consequence to the plant?
And they have generated huge data regarding the fungal community in different parts of the plant. Do they have a relation with its medicinal potential? The authors should discuss more about this.

Validity of the findings

The description of the fungal community of Paris polyphylla Sm. However, I think that the manuscript is merely descriptive and needs major improvements regarding its hypothesis and the discussion of the findings.

---

## Round 0.3 · accepted · Accept

You have met the suggestions from the reviewers.